# Economic, cultural, and social inequalities in potentially inappropriate medication: A nationwide survey- and register-based study in Denmark

Amanda Paust[1,2]*, Claus Vestergaard[1], Susan M. Smith[3], Karina Friis[4], Stine Schramm[5], Flemming Bro[1,2], Anna Mygind[1], Nynne Bech Utoft[1], James Larkin[6], Anders Prior[1]

1 Research Unit for General Practice, Aarhus, Denmark, 2 Department of Public Health, Aarhus University, Aarhus, Denmark, 3 Discipline of Public Health and Primary Care, School of Medicine, Trinity College Dublin, Dublin, Ireland, 4 DEFACTUM, Central Denmark Region, Aarhus, Denmark, 5 National Institute of Public Health, University of Southern Denmark, Copenhagen, Denmark, 6 Department of General Practice, RCSI University of Medicine and Health Sciences, Dublin, Ireland

* amasa@ph.au.dk

## Abstract

### Background

Potentially inappropriate medication (PIM) is associated with negative health outcomes and can serve as an indicator of treatment quality. Previous studies have identified social inequality in treatment but often relied on narrow understandings of social position or failed to account for mediation by differential disease risk among social groups. Understanding how social position influences PIM exposure is crucial for improving the targeting of treatment quality and addressing health disparities. This study investigates the association between social position and PIM, considering the mediation effect of long-term conditions.

### Methods and findings

This cross-sectional study utilized data from the 2017 Danish National Health Survey, including 177,495 individuals aged 18 or older. Data were linked to national registers on individual-level.

PIM was defined from the STOPP/START criteria and social position was assessed through indicators of economic, cultural, and social capital (from Bourdieu's Capital Theory). We analyzed odds ratios (ORs) and prevalence proportion differences (PPDs) for PIM using logistic regression, negative binomial regression, and generalized structural equation modeling. The models were adjusted for age and sex and analyzed separately for indicators of under- (START) and overtreatment (STOPP). The mediation analysis was conducted to separate direct and indirect effects via long-term conditions. Overall, 14.7% of participants were exposed to one or more PIMs, with START PIMs being more prevalent (12.5%) than STOPP PIMs (3.1%). All variables for social position except health education were associated with PIM in a dose-response pattern. Individuals with lower wealth (OR: 1.85 [95% CI

**Data Availability Statement:** The data supporting this study's findings can be made available to authorized research institutions with collaborative agreements that grant access to the Danish registers through Statistics Denmark and the Danish Health Data Authority. The Danish National

Health Survey data was obtained from a third party upon application to the National Steering Group. All interested researchers can apply for this data from the National Institute of Public Health, University of Southern Denmark. More information on access to data is available from Statistics Denmark [accessed on 2024 07-22]: https://www.dst.dk/en/TilSalg/Forskningsservice/Dataadgang.

**Funding:** APa received research grants from the Research Foundation for General Practice (EMN-2018-02975/160-822307, link: https://rltn.dk/fonde/praksisfondene/fonden-for-almen-praksis/), and the Graduate School of Health at Aarhus University (No. 160-780513, link: https://phd.health.au.dk/application/how-to-finance-a-phd). The funding was awarded after peer review of a research proposal. Subsequently, the funders played no role in the analysis, presentation, or interpretation of study results. The Danish National Health Survey was funded by The Capital Region, Region Zealand, The South Denmark Region, The Central Denmark Region, The North Denmark Region, Ministry of Health and the National Institute of Public Health, University of Southern Denmark.

**Competing interests:** The authors have declared that no competing interests exist.

**Abbreviations:** DAG, directed acyclic graph; GP, general practitioner; OR, odds ratio; PIM, potentially inappropriate medication; PPD, prevalence proportion difference; SD, standard deviation.

1.77, 1.94]), lower income (OR: 1.78 [95% CI 1.69, 1.87]), and lower education level (OR: 1.66 [95% CI 1.56, 1.76]) exhibited the strongest associations with PIM. Similar associations were observed for immigrants, people with low social support, and people with limited social networks. The association with PIM remained significant for most variables after accounting for mediation by long-term conditions. The disparities were predominantly related to over-treatment and did not relate to the number of PIMs. The study's main limitation is the risk of reverse causation due to the complex nature of social position and medical treatment.

## Conclusions

The findings highlight significant social inequalities in PIM exposure, driven by both economic, cultural, and social capital despite a universal healthcare system. Understanding the social determinants of PIM can inform policies to reduce inappropriate medication use and improve healthcare quality and equity.

## Author summary

### Why was this study done?

- Potentially inappropriate medication (PIM) is linked to adverse health outcomes and indicates treatment quality issues.

- Previous research has identified social inequalities in medical treatment but often relied on narrow definitions of social position and did not fully account for the mediation effect of long-term conditions.

- There was a need to understand how broader aspects of social position, including economic, cultural, and social capital, influence PIM to inform policy and practice.

### What did the researchers do and find?

- This study utilized data from the 2017 Danish National Health Survey, including 177,495 individuals aged 18 and older, linked with national registers.

- All variables for social position except health education were associated with PIM. Wealth, income, and education level exhibited the strongest associations with PIM, but similar associations were observed for immigrants, not living with other adults, low social support, and limited social networks.

- The association with PIM remained significant for most variables after accounting for mediation by long-term conditions. The disparities were predominantly related to over-treatment and did not relate to the number of PIMs.

**What do these findings mean?**

- The study highlights significant social inequalities in PIM exposure, suggesting that socioeconomic disparities in healthcare persist even in a universal healthcare system. Understanding these disparities can guide efforts to reduce inappropriate medication use and improve patient safety

- The findings indicate that economic, cultural, and social capital are crucial determinants of treatment quality, with economic capital showing the strongest association.

- The study's main limitation is the risk of reverse causation due to the complex nature of social position and medical treatment.

## Introduction

The use of medicines has increased worldwide due to an aging population and a higher prevalence of multimorbidity ($\geq$2 long-term conditions). Half of the world population above age 60 years have multimorbidity, and although multiple medicines may be necessary, an increasing number of medications also increases the risk of adverse drug interactions and suboptimal medical treatment [1,2]. Potentially inappropriate medication (PIM) is often used as an indicator of medical treatment quality. It refers to the omission of appropriate medicines or the use of medicines that may pose more harm than benefit or may have safer alternatives [3]. PIM is associated with a higher risk of emergency visits, medication-related hospitalizations, lower quality of life, increased mortality, and substantial costs for individuals and society [4].

Additionally, social inequality in health is increasing [5]. This escalation is partly attributed to an uneven risk of suboptimal treatment between people with different positions in the social hierarchy (i.e., social positions) [6,7]. PIM serves as a critical measure of inequality in treatment. An inverse association between social position and PIM has already been established [8,9]. Yet, when investigating the connection, studies tend to ignore the differential disease risk between social groups, which mediates much of the association between social position and PIM [9]. Furthermore, investigations often fail to provide a nuanced reflection of the association, e.g., by using inadequate measures of social position [10], possibly due to limited data accessibility or few theoretical considerations [11]. For example, even though income may poorly reflect the economic situation in old age, accumulated wealth or similar measures are rarely used to capture social position [12]. Considering the mediating role of long-term conditions and using a theoretical lens and alternative data sources to define and operationalize social position may allow for a more exhaustive comprehension of how social position relates to suboptimal treatment, which can inform future research, interventions, and policy development.

A highly acknowledged theoretical framework for exploring social position is provided by Bourdieu [10,13]. According to this, social position is shaped by an individual's access to resources (economic, cultural, and social capital) and how these resources are employed in different social fields [14–18]. Thus, the association between social position and PIM could be related to different forms of capital. Moreover, Denmark has extensive national registers and surveys covering various health-related and social aspects. The civil registration number allows individual-level linkage between these data. The theoretical approach combined with the

extensive data available in Denmark can be used to comprehensively examine the relationship between social position, medical treatment, and long-term conditions [19].

Hence, this study aimed to investigate the association between selected indicators of economic, cultural, and social capital with exposure to PIM, using Bourdieu's capital theory and comprehensive national data, and considering the mediation by long-term conditions. To our knowledge, this is the first study to comprehensively explore these associations between social position and PIM. We hypothesized that all forms of capital would be inversely associated with the risk of receiving PIM and that multimorbidity would mediate some of the association.

## Methods

### Design

We performed a cross-sectional study based on information from the Danish National Health Survey and Danish National registers [19,20]. The RECORD criteria were used to conduct and report the study and are reported in S1 Table [21]. The study was based on a prospective protocol. Paust A. Social Inequality in Medical Treatment. protocols.io. 2024. doi: dx.doi.org/10.17504/protocols.io.ewov19wy2lr2/v1.

### Population and setting

We included individuals aged ≥18 years who participated in the Danish National Health Survey 2017; this sample is known to constitute a nationally representative sample of the Danish population [20]. The invited population was selected by Statistics Denmark as a random sample of the adult Danish population for the survey year based on an algorithm to ensure geographic and demographic representativeness [20]. The authors did not have access to the database population used to create the study population. The questionnaire was sent to 321,349 individuals. A total of 183,372 individuals participated; 177,495 were eligible for inclusion (aged ≥18 years). The questionnaire data was collected from January 1, 2017 to May 19, 2017. The register data on demographics, morbidity, and social position [19] was collected on January 1, 2017. The register data on PIM was collected on May 19, 2017. Each variable's response rates and missing data are reported in S2 and S3 Tables. Non-response analysis and data cleaning methods in the study population are elaborated elsewhere [22].

### Data sources

Data on cohabitation, social network, and social support was collected from the Danish National Health Survey 2017 [20]. All remaining information was obtained from Danish registers [19]. The Danish Civil Registration System provided information on sex, age, immigration status, and mortality. Statistics Denmark provided data on education, income, and wealth. The Danish National Patient Register provided information on hospital diagnoses and procedure codes, and psychiatric hospital diagnoses were acquired from the Danish Psychiatric Central Research Register. The Danish National Prescription Register provided data on redemption dates and volumes of prescribed medications. Linking between patients and their general practice clinic was established through the Danish Patient List Database. All information was obtained at individual-level. This was pseudonymized and linked through the Danish personal identification number.

### Outcome variables

**Potentially inappropriate medication.** PIM was defined from the STOPP/START criteria for inappropriate prescribing, developed by O'Mahony and colleagues [23], although

modified to suit the Danish registers and the broader adult population [19]. The criteria included 29 indications for reducing or stopping medication (STOPP) and 10 indications for medication initiation (START) based on risk of drug–drug and drug–disease interactions [24]. The number of PIMs was calculated for all participants on May 19, 2017. The operationalization of PIM is described in S4 Table, and more details on methods can be found elsewhere [24,25]. Specific criteria were set for being identified as at risk for PIM dependent on the criteria in question (e.g., dependent on a specific condition or combination of conditions) [24].

## Exposure variables

According to Bourdieu, social position can be divided into economic, cultural, and social capital [10]. These capital forms were operationalized from 8 indicators (Table 1). Detailed descriptions and argumentations for each indicator are provided in S5 Table. The relationship between variables is illustrated using a directed acyclic graph (DAG) [26] and presented in S1 Fig.

**Economic capital.** We used register-based measures of aggregated household net wealth (quintile categories) and equivalent disposable household income (quintile categories) to describe economic capital. Wealth included the complete portfolio information, i.e., the value of bonds, stocks, cash in banks, real estate, mortgage loans, and the sum of other loans (excluding pension savings).

**Cultural capital.** To capture cultural capital, we employed 3 register-based measures: the highest attained education level aggregated for the household (primary and lower secondary, upper secondary, tertiary/bachelor/equivalent, or master/doctoral/equivalent), having a healthcare-related education regardless of the level (yes, no), and immigration status (Danish origin, immigrant, descendant).

**Social capital.** To capture social capital, we employed 3 survey items: a combined measure of social network, i.e., interaction with friends, neighbors, colleagues, or family outside the household (ranging from infrequent to frequent social contact), cohabitation (living with adult(s) ($\geq$ age 16), not living with adult(s) ($\geq$ age 16)), and social support, i.e., having someone to support you or discuss problems with (always, mostly, sometimes, never, or almost never).

**Table 1. Operationalization of social position.**

| Social position | Indicator | Measures |
|---|---|---|
| Economic capital | Wealth quintile categories | Aggregated household net wealth. |
| | Income quintile categories | Equivalent disposable household income. |
| Cultural capital | Household education level | Highest attained education level aggregated for household. |
| | Healthcare-related education | Having a healthcare-related education (regardless of level). |
| | Immigration status | Immigrant if both parents are foreign citizens or if born abroad. Descendant if at least one parent is immigrant. |
| Social capital | Social network | Interaction with friends, neighbors, colleagues, or family outside household. |
| | Cohabitation | Yes covers living with adults ($\geq$ age 16), no covers not living with adult ($\geq$ age 16). |
| | Social support | Having someone to support you or discuss problems with. |

Operationalization elaborated in S5 Table.

**Covariates.** We included 3 register-based covariates, i.e., sex (male, female), age (using restricted cubic spline variables with 5 knots), and the number of long-term conditions (ranging from 0 to ≥5), as these are associated with both social position and PIM [27]. The Danish Multimorbidity Index was used to identify long-term conditions [28]. These conditions were defined as 39 physical and mental long-term diagnoses (see S6 Table) and identified from Danish registers [19].

## Statistical analyses

For all independent variables, we calculated the count, the means, and the standard deviations for the number of START and STOPP PIMs in the population and estimated the prevalence proportion difference (PPD) adjusted for age and sex. Prevalence proportions were calculated as the number of PIMs per person; thereby, the numerator may exceed one.

To analyze the odds ratios (ORs) and the prevalence proportion ratios of PIM for indicators of social position, we employed logistic regression (logit) and negative binomial regression models (nbreg) and used generalized structural equation modeling (gsem) for mediation analyses. These analyses were conducted as complete case analyses using STATA software version 18. Model 1a used logit to estimate the OR between each indicator of social position and any PIM. Model 1b used nbreg to estimate the prevalence proportion ratios between social position and the number of PIMs, given that the individuals have at least one PIM. Both models were adjusted for age and sex. Model 2 presented the association between social position and PIM (i.e., 1a) stratified by START (2a) PIM and STOPP (2b) PIM, adjusted for age and sex. These models were also stratified by sex in model 3a and 3b. Model 4 examined the direct association between the indicators of social position and PIM (4a) and STOPP PIM (4b) (unmediated by long-term conditions) adjusted for age and sex. The mediation analysis was performed using gsem, separating the direct association from the association mediated by the number of long-term conditions (as illustrated in S1 Fig). This model was chosen to reduce the risk of introducing potential bias from adjusting for a mediator.

For all regression analyses, we included cluster-robust variance estimation to account for non-independence of observations due to participants sharing the same general practitioner (GP), i.e., clustering on GP level. This also considered geographical clustering of participants, as individuals sharing a GP generally share geographical location. Before conducting the analysis, we checked for possible collinearity between exposure variables and only saw substantial collinearity between indicators within the same capital form, e.g., wealth and income. Yet, variables from the same capital form did not enter the same analysis as all 8 indicators were analyzed separately. A brief study protocol informed the study, including hypothesis, main analysis, and specifications for variables, but amendments were made after internal and external review, adding post hoc analysis to the study. These include the mediation analysis (from the inclusion of DAGs based on internal review) and stratified analysis (suggested in external review). Besides, the study was planned as a 1-year follow-up study but was conducted as a cross-sectional study due to the risk of time-dependent bias.

## Ethical considerations

The introductory letter for the survey underscored that participation was voluntary. Hence, upon completion of the questionnaire, respondents provided written consent to engage in the survey. Approval for the survey was obtained from the Danish Data Protection Agency. According to Danish law, this study could not be considered for ethical approval as it did not include human biological material [29]. The study adheres to the principles outlined in the Declaration of Helsinki [30].

## Results

### Descriptive data

The study sample consisted of 177,495 individuals with an average age of 53.1 years and 33.0% diagnosed with 2 or more long-term conditions (Table 2). Overall, 26,252 individuals in the study population were subjected to 1 or more PIMs at the time of the data collection, equivalent to 14.7% of the population. In total, 9.5% had 1 PIM, 3.0% had 2 PIMs, and 2.1% had 3 or more PIMs. Exposure to START PIMs (22,140/177,495 = 12.5%) was more common than exposure to STOPP PIMs (5,555/177,495 = 3.1%). Overall, 92.5% were of Danish origin, 86.2% had more than lower secondary education as the highest attained education level, and 7.8% had a healthcare-related education. In total, 33.6% of the study population reported having a limited social network, defined as having below-average frequency of social contact with people outside the household. In total, 20.9% lived alone or with child(ren) below age 16 years, and 13.0% reported low levels of social support, measured by never or rarely having someone to talk to when in need.

### PIM and adjusted prevalence proportion difference

As demonstrated in Table 3, the mean number of PIM ranged from 0.12 (standard deviation [SD] 0.44) to 0.38 (SD 0.79) PIMs across groups with different economic, cultural, and social capital. While the mean number of START PIM was 0.19 (SD 0.58), the study population was exposed to 0.03 STOPP PIM (SD 0.19). When analyzing the age- and sex-adjusted PIM prevalence proportion, we found that the differences between those with the lowest and highest economic, cultural, and social capital amounted to 1 to 12 additional PIMs per 100 individuals for indicators of economic, cultural, and social capital; the least educated compared to the highest (PPD 0.09 [95% CI 0.07, 0.10]), the poorest compared to the wealthiest (PPD 0.12 [95% CI 0.11, 0.13]), those having least contact with others compared to those having the most (PPD 0.08 [95% CI 0.06, 0.09]), and individuals reporting no one to talk to compared to those reporting often having someone to talk to (PPD 0.08 [95% CI 0.06, 0.09]). The PPDs were lower, but still statistically significant, when comparing immigrants to individuals of Danish origin (PPD 0.04 [95% CI 0.02, 0.06]), and individuals living with other adults to those living alone or with younger children (PPD 0.03 [95% CI 0.03, 1.04]). Those with healthcare education did not have a significantly higher prevalence proportion difference compared to individuals without (PPD 0.01 [95% CI 0.00, 0.02]).

### Association between PIM and economic, cultural, and social capital

When analyzing the ORs, we found that all variables for social position except health education were associated with PIM with a dose-response pattern after adjusting for age and sex (Fig 1, model 1a).

**Economic, cultural, and social capital.** An inverse dose-response association was seen between economic capital and PIM. The groups with the least economic capital had 85% higher odds for PIM compared to the group with the most economic capital (wealth OR: 1.85 [95% CI 1.77, 1.94], income OR: 1.78 [95% CI 1.69, 1.87]) after adjusting for age and sex (Fig 1, model 1a). Among the indicators of cultural capital, the strongest dose-response association with PIM was seen for the highest attained education level in the household (OR: 1.66 [95% CI 1.56, 1.76] for primary/lower secondary school compared to master/doctoral) (Fig 1, model 1a). Social capital had the least substantial association with the odds for PIM compared to the other types of capital. Among the indicators of social capital, social support and social network demonstrated similar dose-response associations with PIM when adjusted for age and sex

**Table 2. Baseline characteristics of the total population and population exposed to PIM.**

| Characteristics | | | All, n (%) | START, n (%) | STOPP, n (%) |
|---|---|---|---|---|---|
| | | | N = 177,495 | N = 22,140 | N = 5,555 |
| Covariates | Sex | Female | 95,672 (53.9) | 9,825 (44.4) | 2,977 (53.6) |
| | | Male | 81,823 (46.1) | 12,315 (55.6) | 2,578 (46.4) |
| | Age | 18–29 years | 22,841 (12.9) | 1,436 (6.5) | 80 (1.4) |
| | | 30–39 years | 20,570 (11.6) | 845 (3.8) | 189 (3.4) |
| | | 40–49 years | 28,891 (16.3) | 1,435 (6.5) | 490 (8.8) |
| | | 50–59 years | 34,940 (19.7) | 3,012 (13.6) | 1,074 (19.3) |
| | | 60–69 years | 33,684 (19.0) | 5,135 (23.2) | 1,380 (24.8) |
| | | 70–79 years | 26,545 (15.0) | 6,465 (29.2) | 1,469 (26.4) |
| | | ≥80 years | 10,024 (5.6) | 3,812 (17.2) | 873 (15.7) |
| | Number of long-term conditions | 0 conditions | 80,963 (45.6) | 2,408 (10.9) | 273 (4.9) |
| | | 1 condition | 37,971 (21.4) | 3,918 (17.7) | 740 (13.3) |
| | | 2 conditions | 23,110 (13.0) | 4,131 (18.7) | 933 (16.8) |
| | | 3 conditions | 15,264 (8.6) | 3,765 (17.0) | 1,018 (18.3) |
| | | 4 conditions | 9,363 (5.3) | 2,948 (13.3) | 923 (16.6) |
| | | ≥5 conditions | 10,824 (6.1) | 4,970 (22.4) | 1,668 (30.0) |
| Outcome | Number of PIMs | 0 | 151,402 (85.3) | - | - |
| | | 1 | 16,887 (9.5) | 14,171 (64.0) | 5,160 (92.9) |
| | | 2 | 5,397 (3.0) | 4,651 (21.0) | 368 (6.6) |
| | | 3 | 3,096 (1.7) | 3,318 (15.0)* | 27 (0.5)* |
| | | 4 | 584 (0.3) | | |
| | | ≥5 | 129 (0.1) | | |
| Economic capital | Wealth quintile categories | 1 (least) | 35,453 (20.0) | 3,916 (17.7) | 1,244 (22.4) |
| | | 2 | 35,453 (20.0) | 4,369 (19.7) | 1,176 (21.2) |
| | | 3 | 35,453 (20.0) | 3,914 (17.7) | 1,055 (19.0) |
| | | 4 | 35,453 (20.0) | 4,554 (20.6) | 1,021 (18.4) |
| | | 5 (most) | 35,453 (20.0) | 5,376 (24.3) | 1,057 (19.0) |
| | Income quintile categories | 1 (least) | 35,483 (20.0) | 6,043 (27.3) | 1,623 (29.2) |
| | | 2 | 35,483 (20.0) | 5,671 (25.6) | 1,459 (26.3) |
| | | 3 | 35,483 (20.0) | 4,020 (18.2) | 951 (17.1) |
| | | 4 | 35,483 (20.0) | 3,333 (15.1) | 812 (14.6) |
| | | 5 (most) | 35,482 (20.0) | 3,073 (13.9) | 710 (12.8) |
| Cultural capital | Immigration status | Immigrant | 1,406 (0.8) | 123 (0.6) | 17 (0.3) |
| | | Descendant | 11,946 (6.7) | 1,057 (4.8) | 233 (4.2) |
| | | Danish origin | 164,079 (92.5) | 20,960 (94.7) | 5,305 (95.5) |
| | Household education level | Primary and lower secondary | 24,428 (13.8) | 4,831 (22.0) | 1,453 (26.4) |
| | | Upper secondary | 78,076 (44.3) | 10,191 (46.5) | 2,540 (46.2) |
| | | Tertiary/bachelor | 51,282 (29.1) | 5,176 (23.6) | 1,134 (20.6) |
| | | Master/doctoral | 22,647 (12.8) | 1,725 (7.9) | 375 (6.8) |
| | Healthcare education | No | 122,117 (92.0) | 13,677 (63.0) | 3,158 (58.0) |
| | | Yes | 10,576 (7.8) | 1,012 (4.7) | 306 (5.6) |

(*Continued*)

**Table 2.** (Continued)

| Characteristics | | | All, _n_ (%) | START, _n_ (%) | STOPP, _n_ (%) |
|---|---|---|---|---|---|
| | | | _N_ = 177,495 | _N_ = 22,140 | _N_ = 5,555 |
| **Social capital** | Social network | 1 (infrequent) | 7,634 (4.6) | 1,234 (6.0) | 406 (8.0) |
| | | 2 | 47,961 (29.0) | 6,232 (30.4) | 1,600 (31.7) |
| | | 3 | 80,619 (48.7) | 9,900 (48.3) | 2,349 (46.5) |
| | | 4 (frequent) | 29,221 (17.7) | 3,116 (15.2) | 692 (13.7) |
| | Cohabitation | No | 34,291 (20.9) | 5,480 (26.8) | 1,589 (31.3) |
| | | Yes | 130,110 (79.1) | 14,991 (73.2) | 3,488 (68.7) |
| | Social support | Never or almost never | 6,973 (4.2) | 1,121 (5.4) | 330 (6.3) |
| | | Sometimes | 14,774 (8.8) | 1,918 (9.2) | 597 (11.5) |
| | | Mostly | 41,864 (25.1) | 5,188 (24.8) | 1,340 (25.8) |
| | | Always | 103,495 (61.9) | 12,722 (60.7) | 2,935 (56.4) |

\*≥3 PIM

PIM, potentially inappropriate medication.

(never versus often and least versus most OR: 1.35 [95% CI 1.26, 1.44 and 1.26, 1.45]) (Fig 1, model 1a). The association between indicators of social position and the number of PIMs, given that individuals received at least 1 PIM, was not clinically relevant (Fig 1, model 1b).

**Undertreatment and versus overtreatment, males versus females.** Exploring the odds ratio for receiving PIM between different indicators of low social position, stratified by STOPP and START PIM, we found that STOPP PIM drove the overall associations. The associations were similar but stronger for STOPP PIM compared to the combined measure for PIM (Fig 2, model 2b versus Fig 1, model 1a). For START PIM (Fig 2, model 2a), a weak association was seen, besides slightly increased odds among descendants compared to individuals of Danish origin (OR: 1.25 [95% CI 1.12, 1.42]) and adverse association for education level and income with odds for START PIM (education OR: 0.79 [95% CI 0.71, 0.88] primary/lower secondary school compared to master/doctoral). This indicates that the odds for undertreatment may increase with increased education level.

Stratifying the START/STOPP analysis (Fig 2) by sex, we found that the capital form most dominant in STOPP PIM association differed between men and women (Fig 3). While men demonstrated a stronger association for economic capital (Fig 3, model 3b), women had a stronger association for cultural and social capital (Fig 3, model 3b). Interestingly, having a healthcare-related education, which has been insignificant in all other analyses, appeared to be strongly associated with STOPP PIM among men (OR: 0.58 [CI 95% 0.45, 0.76]).

**Mediation by long-term conditions.** When accounting for the differential disease risk (i.e., the indirect association mediated by long-term conditions) on the association between social position and PIM, we found that the associations were similar but attenuated both when analyzing PIM in general but also for STOPP PIM (Fig 4, model 4a and 4b). Overall, more than half of the association between PIM and social position was explained by differential disease risk for the most predominant associations, e.g., wealth, income, education, social network, cohabitation, and social support (Fig 3).

## Discussion

Our results showed that low economic, cultural, and social capital were associated with exposure to PIM. All investigated variables for social position, except for having a healthcare

**Table 3. PIM prevalence and adjusted prevalence proportion difference.**

| Characteristics | | Any PIM | | | | START PIM | | | | STOPP PIM | | | |
|---|---|---|---|---|---|---|---|---|---|---|---|---|---|
| | | Count | Mean | SD | Adj. PPD (95% CI) | Count | Mean | SD | Adj. PPD (95% CI) | Count | Mean | SD | Adj. PDD (95% CI) |
| **All** | | **39,970** | **0.23** | **0.63** | | **33,990** | **0.19** | **0.58** | | **5,980** | **0.03** | **0.19** | |
| Wealth quintile categories | 1 (least) | 7,325 | 0.21 | 0.6 | 0.12 (0.11, 0.13) | 5,991 | 0.17 | 0.55 | 0.03 (0.00, 0.07) | 1,334 | 0.04 | 0.2 | 0.03 (0.03, 0.03) |
| | 2 | 7,820 | 0.22 | 0.62 | 0.09 (0.08, 0.10) | 6,521 | 0.18 | 0.56 | 0.01 (-0.02, 0.04) | 1,299 | 0.04 | 0.21 | 0.02 (0.02, 0.03) |
| | 3 | 7,132 | 0.2 | 0.6 | 0.05 (0.04, 0.06) | 6,016 | 0.17 | 0.55 | -0.02 (-0.06, 0.01) | 1,116 | 0.03 | 0.18 | 0.01 (0.01, 0.02) |
| | 4 | 8,161 | 0.23 | 0.64 | 0.03 (0.02, 0.04) | 7,065 | 0.2 | 0.6 | -0.01 (-0.04, 0.02) | 1,096 | 0.03 | 0.19 | 0.01 (0.00, 0.01) |
| | 5 (most) | 9,516 | 0.27 | 0.68 | 0 (Reference) | 8,383 | 0.24 | 0.64 | 0 (Reference) | 1,133 | 0.03 | 0.19 | 0 (Reference) |
| Income quintile categories | 1 (least) | 11,070 | 0.31 | 0.73 | 0.10 (0.09, 0.11) | 9,300 | 0.26 | 0.67 | 0.00 (−0.03, 0.04) | 1,770 | 0.05 | 0.24 | 0.03 (0.02, 0.03) |
| | 2 | 10,344 | 0.29 | 0.71 | 0.08 (0.07, 0.08) | 8,750 | 0.25 | 0.65 | 0.00 (−0.04, 0.03) | 1,594 | 0.04 | 0.23 | 0.02 (0.02, 0.02) |
| | 3 | 7,079 | 0.2 | 0.59 | 0.04 (0.03, 0.05) | 6,071 | 0.17 | 0.55 | 0.00 (−0.04, 0.04) | 1,008 | 0.03 | 0.18 | 0.01 (0.01, 0.01) |
| | 4 | 6,018 | 0.17 | 0.55 | 0.03 (0.02, 0.03) | 5,157 | 0.15 | 0.52 | 0.03 (−0.01, 0.07) | 861 | 0.02 | 0.16 | 0.00 (0.00, 0.01) |
| | 5 (most) | 5,459 | 0.15 | 0.52 | 0 (Reference) | 4,712 | 0.13 | 0.49 | 0 (Reference) | 747 | 0.02 | 0.15 | 0 (Reference) |
| Immigration status | Immigrant | 173 | 0.12 | 0.44 | 0.04 (0.02, 0.06) | 156 | 0.11 | 0.42 | 0.13 (0.00, 0.26) | 17 | 0.01 | 0.11 | 0.00 (0.00, 0.01) |
| | Descendant | 1,942 | 0.16 | 0.55 | 0.00 (−0.01, 0.01) | 1,694 | 0.14 | 0.52 | 0.11 (0.05, 0.16) | 248 | 0.02 | 0.15 | 0.00 (−0.01, 0.00) |
| | Danish origin | 37,855 | 0.23 | 0.64 | 0 (Reference) | 32,140 | 0.2 | 0.59 | 0 (Reference) | 5,715 | 0.03 | 0.2 | 0 (Reference) |
| Household education level | Primary and lower secondary | 9,176 | 0.38 | 0.79 | 0.09 (0.07, 0.10) | 7,596 | 0.31 | 0.73 | −0.02 (−0.07, 0.02) | 1,580 | 0.06 | 0.27 | 0.03 (0.02, 0.03) |
| | Upper secondary | 18,355 | 0.24 | 0.64 | 0.05 (0.04, 0.05) | 15,641 | 0.2 | 0.59 | 0.00 (−0.04, 0.04) | 2,714 | 0.03 | 0.2 | 0.01 (0.01, 0.01) |
| | Tertiary/bachelor | 9,023 | 0.18 | 0.56 | 0.02 (0.01, 0.02) | 7,792 | 0.15 | 0.52 | 0.01 (−0.03, 0.06) | 1,231 | 0.02 | 0.17 | 0.00 (0.00, 0.00) |
| | Master/doctoral | 2,995 | 0.13 | 0.49 | 0 (Reference) | 2,602 | 0.11 | 0.46 | 0 (Reference) | 393 | 0.02 | 0.14 | 0 (Reference) |
| Healthcare education | No | 24,248 | 0.2 | 0.59 | 0.01 (0.00, 0.02) | 20,867 | 0.17 | 0.55 | 0.02 (−0.04, 0.07) | 3,381 | 0.03 | 0.18 | 0.00 (−0.01, 0.00) |
| | Yes | 1,924 | 0.18 | 0.57 | 0 (Reference) | 1,594 | 0.15 | 0.53 | 0 (Reference) | 330 | 0.03 | 0.19 | 0 (Reference) |
| Social network | 1 (infrequent) | 2,411 | 0.32 | 0.75 | 0.08 (0.06, 0.09) | 1,963 | 0.26 | 0.68 | 0.04 (−0.01, 0.10) | 448 | 0.06 | 0.26 | 0.03 (0.02, 0.03) |
| | 2 | 11,401 | 0.24 | 0.65 | 0.02 (0.01, 0.03) | 9,679 | 0.2 | 0.6 | 0.03 (0.00, 0.06) | 1,722 | 0.04 | 0.2 | 0.00 (0.00, 0.01) |
| | 3 | 17,606 | 0.22 | 0.61 | 0.00 (0.00, 0.01) | 15,090 | 0.19 | 0.57 | 0.02 (−0.02, 0.05) | 2,516 | 0.03 | 0.19 | 0.00 (0.00, 0.00) |
| | 4 (frequent) | 5,417 | 0.19 | 0.57 | 0 (Reference) | 4,669 | 0.16 | 0.53 | 0 (Reference) | 748 | 0.03 | 0.17 | 0 (Reference) |
| Cohabitation | No | 10,507 | 0.31 | 0.74 | 0.03 (0.03, 0.04) | 8,779 | 0.26 | 0.68 | 0.03 (0.00, 0.05) | 1,728 | 0.05 | 0.24 | 0.01 (0.01, 0.02) |
| | Yes | 26,404 | 0.2 | 0.59 | 0 (Reference) | 22,665 | 0.17 | 0.55 | 0 (Reference) | 3,739 | 0.03 | 0.18 | 0 (Reference) |
| Social support | Never or almost never | 2,136 | 0.31 | 0.75 | 0.08 (0.06, 0.09) | 1,770 | 0.25 | 0.68 | 0.07 (0.02, 0.12) | 366 | 0.05 | 0.25 | 0.02 (0.02, 0.03) |
| | Sometimes | 3,716 | 0.25 | 0.67 | 0.06 (0.05, 0.07) | 3,077 | 0.21 | 0.62 | 0.08 (0.03, 0.12) | 639 | 0.04 | 0.22 | 0.02 (0.01, 0.02) |
| | Mostly | 9,571 | 0.23 | 0.64 | 0.02 (0.01, 0.03) | 8,133 | 0.19 | 0.59 | 0.05 (0.02, 0.08) | 1,438 | 0.03 | 0.2 | 0.00 (0.00, 0.01) |
| | Always | 22,356 | 0.22 | 0.61 | 0 (Reference) | 19,196 | 0.19 | 0.57 | 0 (Reference) | 3,160 | 0.03 | 0.19 | 0 (Reference) |

CI, confidence interval; PIM, potentially inappropriate medication; PPD, prevalence proportion differences; SD, standard deviation.

Adjusted for age and sex.

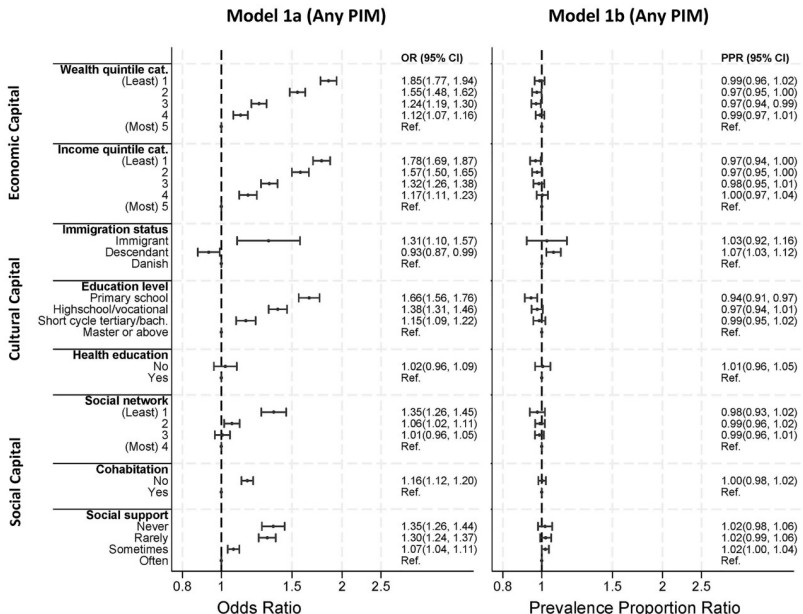

**Fig 1. The OR for exposure to PIM (model 1a) and the prevalence proportion ratio for the exposure to increased number of PIMs (model 1b) between indicators of social position.** Legend text: PIM(s), potentially inappropriate medication(s); OR, odds ratio, PPR, prevalence proportion ratio; Ref., Reference, CI, confidence interval. Adjusted for age and sex.

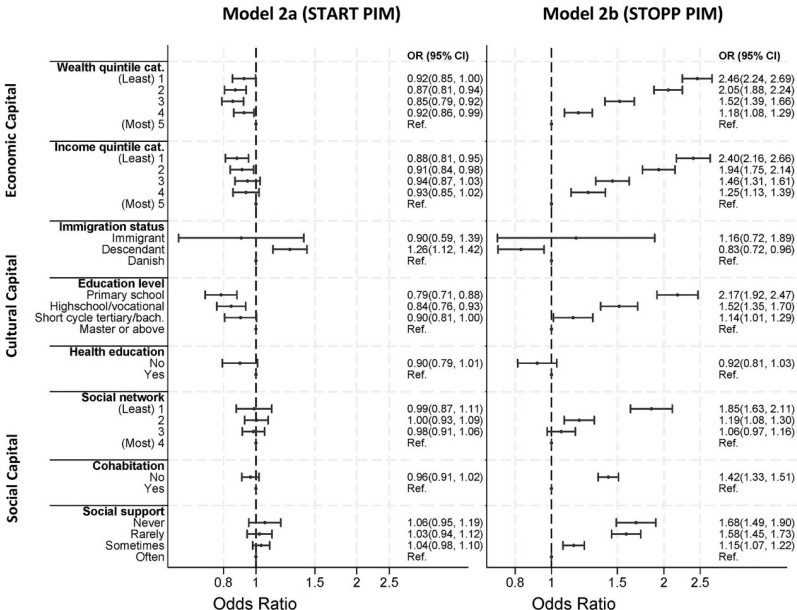

**Fig 2. The OR for exposure to START PIM (model 2a) and STOPP PIM (model 2b) between indicators of social position.** PIM, potentially inappropriate medication; OR, odds ratio; Ref., Reference; CI, confidence interval. Adjusted for age and sex.

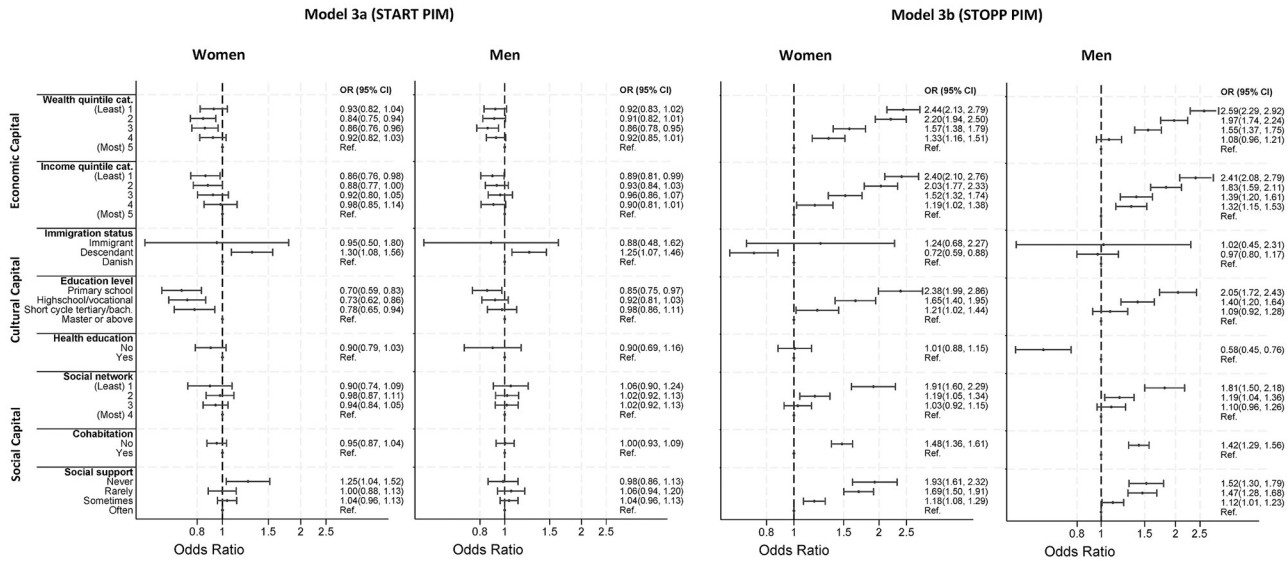

**Fig 3. The sex-stratified odds ratio for exposure to START PIM (model 3a) and STOPP PIM (model 3b) between indicators of social position.** PIM, potentially inappropriate medication; OR, odds ratio; Ref., Reference; CI, confidence Interval. Adjusted for age.

education, demonstrated a significant association with increased PIM after adjustment for sex and age. A dose-response relationship was evident for most variables, indicating that lower social positions corresponded to higher PIM exposure. However, social position was not associated with the number of PIMs. The association between social position and PIM was primarily seen for the indicators of overtreatment (STOPP criteria). The most pronounced

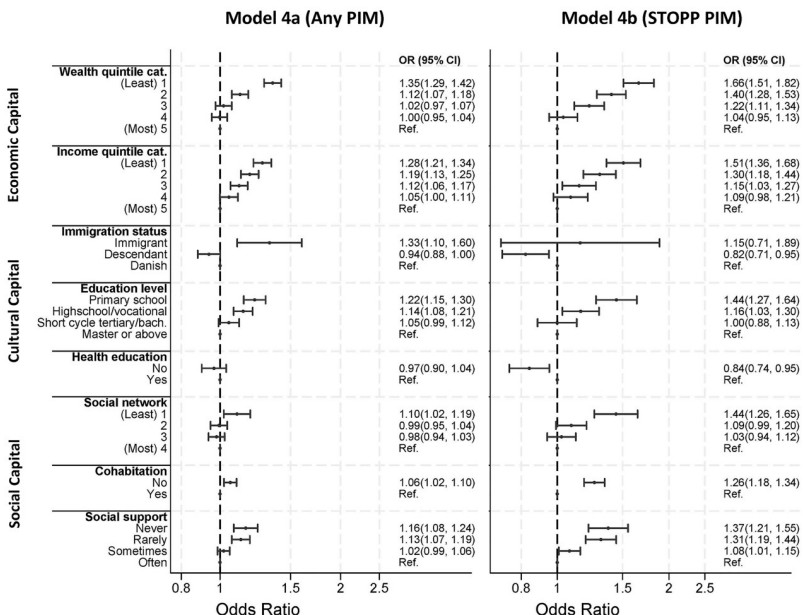

**Fig 4. The odds ratio for exposure to PIM (model 4a) and STOPP PIM (model 4b) between indicators of social position unmediated by long-term conditions.** PIM, potentially inappropriate medication; OR, odds ratio; Ref., Reference; CI, confidence Interval. Adjusted for age and sex.

associations with increased PIM were found for individuals with the lowest levels of income, education, and wealth. Other significant factors included being an immigrant, having low social support, and having a limited social network. The sex-stratified analyses revealed that these associations were largely consistent across both men and women, although the magnitude of associations varied between sexes. While the number of long-term conditions mediated much of this relationship, the association between social position and PIM remained statistically significant for all indicators of social position, excluding healthcare education. These findings suggest social inequality in PIM exposure, persisting even after accounting for disparities in disease risk.

Our findings are consistent with prior research that has established associations between PIM use and various measures of social position, including living alone, income, and education attainment [31–34]. However, our study reveals additional dimensions of social position that may contribute to disparities in medical treatment. Notably, we shed light on the significance of social capital, which is a frequently overlooked factor in epidemiological research on social inequalities, as education, income, and occupation are commonly used to define social position [35,36]. Particularly, low social support was associated with PIM, although much of the association was mediated by long-term conditions. Limited social network and living without other adults had little association with PIM after considering differential disease risk. This may indicate that social capital lies in the ability to utilize the available resources in one's network rather than in the network itself.

This study found that wealth and income exhibited the strongest association with PIM, which seems unexpected within the context of a primarily free-of-charge universal healthcare system. For potentially omitted medicines (START PIMs), this may be explained by the fact that medication expenses are only fully reimbursed when exceeding a certain threshold (590 EUR over one year (2017)). However, our study suggests that potentially inappropriate medicines (STOPP PIMs) are the primary contributors to social inequality in PIM exposure. This indicates that inequality in medical treatment is likely more nuanced than individuals' ability to pay for medicines. Drawing from Bourdieu's work, social inequality is perpetuated through social interactions and power dynamics [37], emphasizing the complex and interconnected nature of the mechanisms behind unequal access to medical treatment.

This study provided an opportunity to explore PIM in the context of a wide range of complex interconnected factors. The epidemiological approach to managing the link between social position, long-term conditions, and PIM, inspired by directed acyclic graphs [26], strengthened the study and improved the accuracy and the ability to draw insightful conclusions. Moreover, the large nationally representative survey population provided information on social capital, and this data was individually linked with national registries to provide reliable data on the population's demographic characteristics, long-term conditions, and social position. The extensive study population enhances the generalizability of the findings to the broader Danish population, potentially also to other countries with similar characteristics, e.g., healthcare and socioeconomic structures. Also, the theory-driven operationalization of social position allowed for a strong foundation for understanding the social processes underlying health inequalities [11].

Nonetheless, social inequality is a complex and entangled issue, and the nuances are difficult to fully capture. Some variables may have acted as mediators or moderators of the association, and the complex nature of the study increased the risk of reverse causation in the study. Moreover, residual confounding from age and long-term conditions may occur; all conditions were weighted equally as we had no data on disease severity. Furthermore, our results were based on drug redemption rather than drug prescription or adherence, preventing us from determining when the issue of PIM arises on the path from prescription to the patient. PIM is

a valid concept on a population level, but for the individual patient, there may be a good reason for prescribing and taking the medication despite the risk of adverse effects. For example, some patients may choose to continue taking a long-term nonsteroidal anti-inflammatory drug to relieve pain despite the potential risk of gastrointestinal and cardiovascular complications. Finally, our study population was based on survey respondents, and varying response rates might have introduced selection bias, potentially affecting the internal and external validity of the results.

Our findings underscore that social inequality in medical treatment remains a critical concern for the quality of care and the safety of medicine use, even in a universal free-of-charge healthcare system, on a national scale and after accounting for age, sex, and differential disease risk. Differences in patient behavior will possibly explain some of the associations. For example, patients with different social positions vary in their capacity to engage in shared decision-making on treatments [38]. However, we must acknowledge that such treatment inequalities could be attributable to the healthcare system and providers, including poor treatment quality, implicit provider biases, limited continuity of care, organizational barriers, or other structural factors [39–41]. Moreover, lowering the healthcare access threshold is crucial to equitable use of healthcare services. Access to healthcare may relate to approachability, acceptability, availability and accommodation, affordability and appropriateness [42]. Addressing unequal medical treatment calls for a nuanced approach in order to provide better treatment for patients with the greatest needs. By acknowledging and addressing the impact of unequal treatment outcomes, we can strive to promote more equitable and effective patient care.

Social inequality in healthcare and treatment is not fully understood. In this study, we used PIM as an indicator of treatment quality. However, further investigations are needed to explore other indicators of treatment quality and the interplay between capitals to understand the underlying differences in the medical treatment of various social groups. For example, methods such as latent class analysis or cluster analysis could assist in identifying hidden groupings in exposure to poor medical treatment. Moreover, valuable insight could be gained from conducting similar research among populations underrepresented in research, e.g., homeless people or undocumented migrants.

This study showed that the individuals' economic, cultural, and social capitals were highly associated with PIM, even after accounting for the disparities attributable to differential disease risk. The disparities were predominantly related to overtreatment rather than undertreatment and did not relate to the number of PIMs. Overall, economic capital exhibited the strongest association with PIM, followed by cultural and social capital. It is necessary to consider the association between PIM and social position when designing interventions and providing services to improve the quality of treatment and patient safety.

## Supporting information

**S1 Table. The RECORD statement.**
(PDF)

**S2 Table. Response rates in the Danish National Health Survey 2017.**
(PDF)

**S3 Table. Missing data.**
(PDF)

**S4 Table. PIM criteria definitions according to the register-adapted STOPP/START criteria.**
(PDF)

**S5 Table. Description of operationalization of capital forms.**
(PDF)

**S6 Table. Long-term conditions in the Danish Multimorbidity Index.**
(PDF)

**S1 Fig. Illustration of variables with directed acyclic graph (DAG).**
(PDF)

## Acknowledgments

We kindly thank the survey participants and Lone Niedziella for proofreading.

The content is solely the authors' responsibility and does not necessarily represent the official views of the funders.

## Author Contributions

**Conceptualization:** Amanda Paust, Stine Schramm, Flemming Bro, Anna Mygind.

**Data curation:** Amanda Paust, Claus Vestergaard, Stine Schramm, Nynne Bech Utoft, Anders Prior.

**Formal analysis:** Amanda Paust, Claus Vestergaard, Susan M. Smith, Karina Friis, Nynne Bech Utoft, Anders Prior.

**Funding acquisition:** Amanda Paust.

**Investigation:** Amanda Paust, Nynne Bech Utoft, Anders Prior.

**Methodology:** Amanda Paust, Claus Vestergaard, Susan M. Smith, Karina Friis, Nynne Bech Utoft, James Larkin, Anders Prior.

**Project administration:** Amanda Paust, Anders Prior.

**Resources:** Amanda Paust, Anders Prior.

**Software:** Amanda Paust, Claus Vestergaard, Nynne Bech Utoft, Anders Prior.

**Supervision:** Amanda Paust, Susan M. Smith, Stine Schramm, Flemming Bro, Anna Mygind, Anders Prior.

**Validation:** Amanda Paust, Claus Vestergaard, Karina Friis, Stine Schramm, Anders Prior.

**Visualization:** Amanda Paust, Claus Vestergaard, Anders Prior.

**Writing – original draft:** Amanda Paust.

**Writing – review & editing:** Amanda Paust, Claus Vestergaard, Susan M. Smith, Karina Friis, Stine Schramm, Flemming Bro, Anna Mygind, Nynne Bech Utoft, James Larkin, Anders Prior.

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
