## [Editor Report · Decision Letter 0]

20 Dec 2023

Dear Dr Paust, 

Thank you for submitting your manuscript entitled "Economic, cultural, and social inequalities in potentially inappropriate medication: a nationwide survey- and register-based study" for consideration by PLOS Medicine.

Your manuscript has now been evaluated by the PLOS Medicine editorial staff and I am writing to let you know that we would like to send your submission out for external peer review.

Please re-submit your manuscript within two working days, i.e. by Dec 22 2023 11:59PM.

Feel free to email me at pdodd@plos.org or the team at plosmedicine@plos.org if you have any queries relating to your submission.

Kind regards,

Pippa

Philippa Dodd, MBBS MRCP PhD

PLOS Medicine

---

## [Decision Letter · Decision Letter 1]

13 Mar 2024

Dear Dr. Paust,

Many thanks for submitting your manuscript "Economic, cultural, and social inequalities in potentially inappropriate medication: a nationwide survey- and register-based study PMEDICINE-D-23-03737R1” to PLOS Medicine. The paper has been reviewed by two subject experts and a statistician; their comments are included below and can also be accessed here: [LINK]

As you will see, the reviewers were very positive about the paper but, they raised a number of questions about specific study details and the methodological approach. After discussing the paper with the editorial team and an academic editor with relevant expertise, I’m pleased to invite you to revise the paper in response to the reviewers’ comments. We plan to send the revised paper to some of all of the original reviewers*, and of course we cannot provide any guarantees at this stage regarding publication.

When you upload your revision, please include a point-by-point response that addresses all of the reviewer and editorial points, indicating the changes made in the manuscript and either an excerpt of the revised text or the location (eg: page and line number) where each change can be found. Please submit a clean version of the paper as the main article file and a version with changes marked should as a marked-up manuscript. Please also check the guidelines for revised papers at http://journals.plos.org/plosmedicine/s/revising-your-manuscript for any that apply to your paper.

We ask that you submit your revision by April 3rd 2024. However, if this deadline is not feasible, please contact me by email, and we can discuss a suitable alternative.

Please don’t hesitate to contact me directly with any questions (pdodd@plos.org). If you reply directly to this message, please be sure to ‘Reply All’ so your message comes directly to my inbox.

Kind regards,

Pippa

Philippa Dodd MBBS MRCP PhD

PLOS Medicine

plosmedicine.org

pdodd@plos.org

*Please note: If your article is accepted, you may have the opportunity to make the peer review history publicly available. The record will include editor decision letters (with reviews) and your responses to reviewer comments. If eligible, we will contact you to opt in or out.

Editorial comments:

1) It took rather a long time to secure reviewers for your manuscript and we thank you for your patience. We agree with the reviewers and with the academic editor that your manuscript could be further strengthened and request that you respond to all comments in full.

2) We agree that stratifying your analyses by sex would be a very valuable and we agree that investigating STOPP and START separately would also add additional insights and nuance. Please amend accordingly.

3) We agree with the statistical reviewer that reporting your study according to RECORD would be appropriate. Please see here for further details https://www.equator-network.org/reporting-guidelines/record/. When completing the checklist, please use section and paragraph numbers, rather than page numbers as these often change in event of publication.

4) For all observational studies we request that in the manuscript text, you please indicate: (1) the specific hypotheses you intended to test, (2) the analytical methods by which you planned to test them, (3) the analyses you actually performed, and (4) when reported analyses differ from those that were planned, transparent explanations for differences that affect the reliability of the study's results. If a reported analysis was performed based on an interesting but unanticipated pattern in the data, please be clear that the analysis was data-driven.

5) Did your study have a prospective protocol or analysis plan? Please state this (either way) early in the Methods section.

Comments from the reviewers:

Reviewer #1: See attachment

Michael Dewey

Reviewer #2: This is an interesting manuscript, aiming to divulge more of the hard to define concept social position and its relation to health inequalities.

The article looks at potentially inappropriate medication (PIM) and social inequality. By using Bourdieu´s capital theory the authors are including not only economic and cultural capital but also social capital. By using individual answers from a household survey in combination with register data they are able to analyse the association between PIM and social inequality on an individual level.

They can confirm previous findings regarding social inequality expressed by economic and cultural capital and these findings are also detected regarding social capital, however weaker.

Despite this, I have a couple of remarks regarding the manuscript:

My main objection is that the results are not reported separately for men and women. Women utilize more health care and consume more medicine and in my mind, this is a fundamental reason to make a gender division.

Minor objections:

In the introduction where the authors outline social class and there will to analyse according to Bourdieu´s work no direct reference is made to his work, only indirect references. Although this reference is made in the supplement, it is appropriate to have it in the main text.

In the first section of the results section (descriptive) a textual briefing of table 2 is made. Unfortunately, the order in the briefing and the table are not equivalent. Better if the orders not are different.

In table 3, please check the count figures for wealth quintiles. It looks as if the order is wrong.

The authors have analysed each variable by its own but it would have been interesting to see some example of interaction, for example can high social capital compensate low economic capital.

All in all, I find this interesting not only for me but for a wider audience if corrections are made.

Reviewer #3: Thank you for the opportunity to review the manuscript. This is an interesting and novel study investigating the association between social position and quality of medication use. The research question and statistical analysis are generally well-defined and straight forward. The main limitation is that the theoretical framework is not sufficiently integrated and that the outcome and some analytical decisions should be further problematized/discussed.

Below, I list some issues that in my view, would improve the paper.

Main comments:

1. I am supportive of the authors decision to use more indicators than education/income/occupation to assess social position. However, I am a bit skeptical about the theoretical connection to Bourdieu's capital terminology. Although I am far from an expert on Bourdieu's work, I have some recommendation to potentially bridge the gap between the analysis and the theoretical discussion: First, the mechanisms underlying the associations as presented in the Discussion should be expanded. Currently, the presented mechanisms are largely materialistic and immediate in nature. In my mind, Bourdieu's work is more concentrated on how social stratification is transferred over time periods/generations and how the capital is used to distinguish/differentiate oneself in relation to other people. It is a bit hard to see how this relates to inappropriate medication use Second, including healthcare-related education as a form of Cultural capital fits well with the study at hand but is more unclear in relation to the term cultural capital in general, and should be better justified.

2. I have some comments relating to the selected outcome: a) Please provide a reference to the STOPP/START criteria, also indicating which version of STOPP/START you used. b) The STOPP/START criteria were developed specifically for older adults, because some medications that are appropriate in other age groups are inappropriate in older adults (e.g. NSAIDs). Hence, you will need to motivate the use of these criteria in other age groups. c) The START criteria are often not included in studies about medication quality, I would be interested to see the results for STOPP and START separately, also as there might be slightly different process generating inequality for the two different domains. d) It is clearly stated that the PIMs were calculated as a one-day point prevalence (which is a good decision especially for drug-drug interactions). However, I can not find information on the method used to calculate the one-day point prevalence, how long was the look-back period, was duration calculated using the DDD-method etc.?

3. The analytical decision to conduct mediation analysis to address long-term conditions should be better supported. I generally agree with the decision, but I think it could be more clearly stated what it is and what the benefits are.

Minor comments:

4. Does reference #5 really support the authors statement?

5. The category 'living alone or with child(ren)' is unclear and should be re-named.

6. I recommend removing "strongly" in the first sentence of the Discussion (Row 226)

[LINK]

Comments from the Academic Editor:

Overall, I think it is quite well done but I think opportunities to make it quite strong. Happy to take a look after reviews are in or after authors respond to them.

This is a well-written article using a robust dataset. Epidemiological analyses are well-thought out and theoretically informed. A few suggestions for improvements to make analyses more robust.

-Consider differences between medications that should have been started but weren’t vs. not stopping a necessary medication. Particularly in terms of socioeconomic and racial disparities, access is substantial driver and will mostly affect whether necessary medication not started. I think I would be more interested in these separately rather than an aggregate measure as I think the mechanisms are likely quite different (quality itself is a broad construct and has many dimensions). Consider reporting each separately rather in aggregate.

-See Levesque et al framework on access. I think this can complement existing theoretical foundation and perhaps fine tune analysis/conceptualization. https://equityhealthj.biomedcentral.com/articles/10.1186/1475-9276-12-18.

-I am confused about adjusted risk differences presented. e.g., for wealth, it seems like mean increases as wealth goes goes up, but this is opposite of adjusted risk difference. Is it true the adjustment fully reversed direction of relationship?

-

I think it would be quite interesting to also consider interactions between different types of capital. E.g., lack of social capital may be particularly exacerbated in the presence of lack of economic capital. High economic capital may counteract impact of low social capital etc.

-I think there is also an element of who your contacts are. This may be captured in the interaction between social and economic capital (e.g., are you connect to well off and well connected individuals?). I imagine the data doesn’t actually capture this directly, but this could provide useful inferences.

-In truth, there may be complex interactions between the different type of capital that essentially identify different subgroups phenotypes. Although complicated to do, using latent class analysis with the different measures of social position to identify these phenotypes and then using this as the exposure would be quite interesting. Could think of other methods for segmentation as well.

-I think would be very helpful to think of ways to consider how these disparities are concentrated. Something like Lorenz curves, Gini index, concentration indices can help to understand how much of the disparities are concentrated at the extremes or whether they are more evenly distributed. Beyond the association, this could help understand where the public health burden really lies.

-What important measurement issues may be present? These are all hard concepts to actually measure and capture and probably worth some discussion.

-The use of DAGs in thinking through analyses for these complex phenomena is a strength. I would include these in the supplement so it is clear how each different analysis was conceptualized. This is important work that the authors have done which should be highlighted.

1. Please upload any figures associated with your paper as individual TIF or EPS files with 300dpi resolution at resubmission; please read our figure guidelines for more information on our requirements: http://journals.plos.org/plosmedicine/s/figures. While revising your submission, please upload your figure files to the PACE digital diagnostic tool, https://pacev2.apexcovantage.com/. PACE helps ensure that figures meet PLOS requirements. To use PACE, you must first register as a user. Then, login and navigate to the UPLOAD tab, where you will find detailed instructions on how to use the tool. If you encounter any issues or have any questions when using PACE, please email us at PLOSMedicine@plos.org.

To submit your revised manuscript please use the following link:

---

## [Decision Letter · Decision Letter 2]

12 Jul 2024

Dear Dr. Paust,

Thank you very much for re-submitting your manuscript "Economic, cultural, and social inequalities in potentially inappropriate medication: a nationwide survey- and register-based study in Denmark" (PMEDICINE-D-23-03737R2) for review by PLOS Medicine.

I have discussed the paper with my colleagues and the academic editor and it was also seen again by 3 reviewers. I am pleased to say that provided the remaining editorial and production issues are dealt with we are planning to accept the paper for publication in the journal.

[LINK]

We look forward to receiving the revised manuscript by Jul 19 2024 11:59PM.   

Kind regards,

Pippa

Philippa Dodd, MBBS MRCP PhD

Senior Editor 

PLOS Medicine

plosmedicine.org

Requests from Editors:

GENERAL

Thank you for your very detailed and considered responses to previous editor and reviewer comments. Please respond to further comments detailed below prior to publication.

Many of the editorial requests pertain to specific content and formatting requirements, some may not apply and others may have already been incorporated into the manuscript but please review each item and maned as necessary.

We agree with the reviewer (please see below) regarding the inclusion of the sex disaggregated data within the main manuscript as opposed to supporting information. Please amend accordingly.

DATA AVAILABILITY STATEMENT

Please include a URL for Statistics Denmark and the Danish Health Data Authority for those wishing to apply for access to data.

ABSTRACT

Please structure your abstract using the PLOS Medicine headings (Background, Methods and Findings, Conclusions).

Please combine the Methods and Findings sections into one section, “Methods and findings”.

Abstract Background: Please provide context of why the study is important. The final sentence should clearly state the study question.

Abstract Background: Please provide context of why the study is important. The final sentence should clearly state the study question.

Abstract Methods and Findings:

Please ensure that all numbers presented in the abstract are present and identical to numbers presented in the main manuscript text.

Please include the study design, population and setting, number of participants, years during which the study took place, length of follow up, and main outcome measures.

Please quantify the main results with 95% CIs and p values.

Please include the important dependent variables that are adjusted for in the analyses.

Please include the actual amounts and/or absolute risk(s) of relevant outcomes (including NNT or NNH where appropriate), not just relative risks or correlation coefficients. (example for absolute risks: PMID: 28399126). 

Please include a summary of adverse events if these were assessed in the study.

In the last sentence of the Abstract Methods and Findings section, please describe the main limitation(s) of the study's methodology.

Abstract Conclusions:

Please address the study implications without overreaching what can be concluded from the data; the phrase "In this study, we observed ..." may be useful.

Please interpret the study based on the results presented in the abstract, emphasizing what is new without overstating your conclusions.

Please avoid vague statements such as "these results have major implications for policy/clinical care". Mention only specific implications substantiated by the results.

Please avoid assertions of primacy ("We report for the first time....")

STATISTICAL REPORTING 

Throughout, including tables and figures, please quantify the main results with 95% CIs and p values.

When reporting p values please report as <0.001 and where higher as p=0.002, for example. If not reporting p values, for the purpose of transparent data reporting, please clearly state the reasons why not. When reporting 95% CIs please separate upper and lower bounds with commas instead of hyphens as the latter can be confused with reporting of negative values.

Please include the actual amounts and/or absolute risk(s) of relevant outcomes (including NNT or NNH where appropriate), not just relative risks or correlation coefficients. (example for absolute risks: PMID: 28399126).

AUTHOR SUMMARY

Thank you for including an Author Summary which reads very nicely.

The author summary should immediately follow the Abstract in your revised manuscript. Please amend.

It would be helpful to provide a couple of examples of the ‘variables for social position’ that your refer to.

In the final bullet point of ‘What Do These Findings Mean?’, please describe the main limitations of the study in non-technical language.

Line 69 – pleas remove the sub-heading ‘Manuscript’.

INTRODUCTION

Please ensure that the introduction addresses past research and explain the need for and potential importance of your study. Indicate whether your study is novel and how you determined that. If there has been a systematic review of the evidence related to your study (or you have conducted one), please refer to and reference that review and indicate whether it supports the need for your study.

METHODS and RESULTS

Please report the number of [patients, samples, etc] and dates of recruitment, and account for all methods used in your study.

Please define "lost to follow-up" as used in this study. Other reasons for exclusion should be defined.

Please define the length of follow up (eg, in mean, SD, and range).

Please provide the actual numbers of events for the outcomes, not just summary statistics or ORs.

Please present numerators and denominators used to derive percentages.

Please ensure to indicate where analyses are adjusted and which factors are adjusted for.

As for the abstract, please ensure to quantify the main results with 95% CIs and p values.

When a p value is given, please specify the statistical test used to determine it.

As above please include the disaggregated data within the main manuscript.

TABLES and FIGURES

As above please include the disaggregated data within the main manuscript.

Please see here for guidelines on submitting and citing figures https://journals.plos.org/plosmedicine/s/figures#loc-how-to-submit-figures-and-captions

Please provide titles and legends for all tables and figures (including those in Supporting Information files).

Please ensure that each table and figure is affiliated to a caption which clearly describes the figure content without the need to refer to the text.

Please ensure that all abbreviations are defined in the caption or an appropriate footnote, including those used to report statistical information.

Please ensure to include the meaning of any dots/lines/bars.

Please consider avoiding the use of red and green in order to make figures more accessible to those with colour blindness.

To help facilitate transparent data reporting, where adjusted analyses are presented please also present the unadjusted analyses for comparison. In a caption or footnote please detail all factors adjusted for.

DISCUSSION

Please ensure to present and organize the Discussion as follows: a short, clear summary of the article's findings; what the study adds to existing research and where and why the results may differ from previous research; strengths and limitations of the study; implications and next steps for research, clinical practice, and/or public policy; one-paragraph conclusion. Please avoid the use of sub-headings such that the discussion reads as continuous prose.

Lines 347, 365 and 367 – please remove these statements from the main manuscript and include only in the manuscript submission form when you resubmit your manuscript. At the time of publication these will be compiled as metadata.

REFERENCES

For in-text reference callouts please place citations in square parentheses separate by commas. For example, [1,3,6] or [1-3]. Please check and amend throughout all sub-sections of the manuscript and supporting files.

In the bibliography please ensure that you list up to but no more than 6 author names followed by et al.

For all web references please ensure you include an, ‘Accessed [date].’

Journal name abbreviations should be those listed in the National Center for Biotechnology Information (NCBI) databases.

SUPPORTING INFORMATION

In the published article, supporting information files are accessed only through a hyperlink attached to the captions. For this reason, you must list captions at the end of your manuscript file. You may include a caption within the supporting information file itself, as long as that caption is also provided in the manuscript file. Do not submit a separate caption file.

As the supporting information files are contained with a single file, please label and cite the ‘supplementary appendices’ and ‘initial project description’ as follows: 

Please label the files as ‘S1 Supplementary appendices’ and ‘S2 Study protocol and analysis plan’ respectively.

Please apply alphabetical labelling to each table and figure contained within the S1 file. For example, ‘Fig A’ to ‘Fig Z’ and ‘Table A’ to ‘Table Z’.

Plain text does not need to be labelled and can just be given a title as necessary. For example, ‘Statistical Analysis Plan’.

Please cite tables/figures as ‘Fig A in S1 Supplementary appendices’ and/or ‘Table A in S1 Supplementary appendices’, for example.

SOCIAL MEDIA

To help us extend the reach of your research, please detail any X (formerly Twitter) handles you wish to be included when we tweet this paper (including your own, your coauthors’, your institution, funder, or lab) in the manuscript submission form when you re-submit the manuscript.

Comments from Reviewers:

Reviewer #1: The authors have addressed all my points

Michael Dewey

Reviewer #2: Thank you for your revised manuscript. You have done a great job of meeting all reviewers' comments and the manuscript has now improved.

However, I do have a continuing remark regarding gender stratification:

I appreciate that the appendix now have separate tables and figures for women and men and, to stress my point, there are differences. Not many, but enough to mention in the results and discussion.

I agree that it is a lot of data but I would have preferred to have the gender stratified tables and figures in the main article, not in the appendix, but I leave this decision to the editor.

Also, maybe the interaction analysis would be too much for the main article, bur since you have performed the analysis, it would have been nice to see the results (gender stratified!) in the appendix. 

Finally, eFigure 2 clarifies the difference between models a and b in the heading (START and STOP). This explanation would benefit the figures in the manuscript.

Reviewer #3: Thank you for the opportunity to review this manuscript again. The authors have done a great job revising the analytical and theoretical part of the manuscript. I have some minor comments that I think would improve an already well-developed manuscript.

Below, I list some issues that in my view, would improve the paper.

Minor comments:

1. The separation of PIMS into STOP and START have improved the paper substantially, I now agree with the analytical decisions. However, the authors should revise how they describe these results in the Result-section and in Discussion (under the heading key-finding) for style and clarity. 

2. Row 327-329. I would argue that both gastrointestinal and cardiovascular complications are more recognized side-effects of long-term NSAID use in older adults. Consider changing.

[LINK]

---

## [Editor Report · Decision Letter 3]

6 Sep 2024

Dear Dr Paust, 

On behalf of my colleagues and the Academic Editor, Dr. Aaloke Mody, I am pleased to inform you that we have agreed to publish your manuscript "Economic, cultural, and social inequalities in potentially inappropriate medication: a nationwide survey- and register-based study in Denmark" (PMEDICINE-D-23-03737R3) in PLOS Medicine.

Prior to publication and at the time you complete your formatting changes as detailed below, please also include the RECORD checklist as supporting information. When completing the checklist please refer to section and paragraph numbers as opposed to page/line numbers as these often change at publication. 

PRESS

Kind regards,

Pippa

Philippa C. Dodd, MBBS MRCP PhD 

Senior Editor 

PLOS Medicine